# GCF: Generalized Causal Forest for Heterogeneous Treatment Effect Estimation using Nonparametric Methods

## Abstract

Heterogeneous treatment effect (HTE) estimation with continuous treatment is essential in multiple disciplines, such as the online marketplace and pharmaceutical industry. The existing machine learning (ML) methods, like forest-based modeling, either work only for discrete treatments or make partially linear or parametric assumptions that may suffer from model misspecification. To alleviate these problems, we extend causal forest (CF) with non-parametric dose-response functions (DRFs) that can be estimated locally using kernel-based Double/Debiased ML estimators. Moreover, we propose a distance-based splitting criterion in the functional space of Partial DRFs to capture the heterogeneity for continuous treatments. We call the proposed algorithm generalized causal forest (GCF) as it generalizes the use case of CF to a much broader setup. We show the effectiveness of GCF compared to SOTA on synthetic data and proprietary real-world data sets.

## 1 Introduction

Heterogeneous treatment effect (HTE) estimation has been of growing interest for decision-makers in a wide spectrum of contexts. It uncovers the effect of interventions at sub-group levels, thereby providing highly tailored suggestions rather than a one-size-fits-all policy. When it comes to precision medicine, HTE provides the leveraged information that physicians need to precisely treat different patients with proper dosages of drugs depending on their genes, living habits, and EHR history. Recently, with the emerging big data that exhibits astronomical complexities, HTE has been essential for online marketplaces (Syrgkanis et al., 2021; Du et al., 2019; Ye et al., 2018). For large-scale data that presents a huge challenge, algorithms that adapt the booming machine learning (ML) techniques to HTE estimation are proposed (Künzel et al., 2019; Zhao et al., 2017; Chernozhukov et al., 2018; Hill, 2011). However, these methods largely focus on binary or discrete treatments and thereby being not widely applicable, given the fact that continuous treatments are prevalent in practice. Examples include targeting customers with continuous incentives, customizing the duration of ads to improve user engagement, and design optimal interest rates to maximize revenue while control for default.

For HTE estimation with continuous treatments, it could be feasible to discretize the treatments and utilize the above approaches, but the limitations are two-phase and bring non-trivial challenges to researchers. First, ordinal treatments like different levels of education cannot be treated as categorical otherwise the order information across treatments will be lost, and secondly, discretizing or bucketing has the limitations of model inaccuracy and cannot recognize the right patterns of the outcome across treatments. To elucidate the complex relationship between treatment and outcome, researchers put forward ideas of dose-response function (DRF).

The estimation of DRF in the presence of high dimensional features can be viewed as a natural extension of regression problems with additional complexities. The success of random forest (RF) for regression problems motivates the development of causal forest (CF) (Athey et al., 2019) and orthogonal random forest (Oprescu et al., 2019) and the modification of Bayesian Additive Random Forest (Woody et al., 2020; Hahn et al., 2020). These methods partition the feature space with a splitting criterion fitted on treatments that can provide accurate estimations for HTE. But often

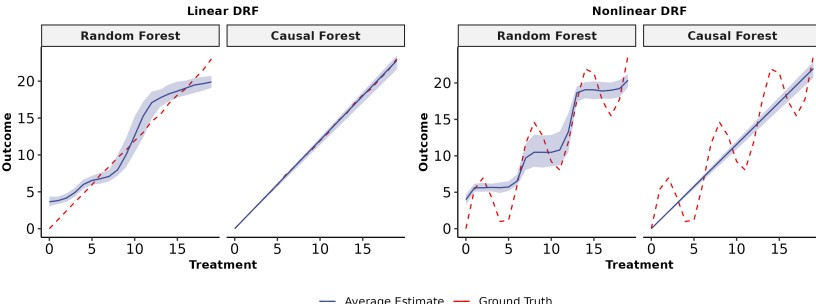

Figure 1: ADRF estimates of random forest and causal forest in both linear and nonlinear DRF specifications. Blue line denotes model average estimations. Confidence bounds of model estimates generated by multiple simulation run are marked in light-blue shade. Ground truth of testing data set is represented in red dash-line.

they posit linear or partially linear models of DRF which are vulnerable to model misspecifications. Figure 1 is an illustration of model misspecification of random forest and CF.

Doubly Robust estimators (DREs) allows for misspecification of either the treatment or outcome regression. Recent work (Kennedy et al., 2017) combines the estimation of generalized propensity score with regression models of the outcome that can further reduce the bias and use kernel regression for non-parametric estimation of DRF. However, it may suffer from the curse of dimensionality when the feature space is high-dimensional given large-scale instances. To advance, (Colangelo & Lee, 2020) provide kernel-based DML estimators that can easily deal with high-dimensional data. Nevertheless, the localization and weighting mechanism of CF is not taken into account, which is essential for capturing heterogeneity. It remains unexplored that how to effectively combine ML methods with localized estimations under no model assumptions.

In this paper, we propose generalized causal forest (GCF) by extending CF to a generalized one with a brand new splitting criterion, and herein the generalization is two-fold. First, we generalize the local linear model of CF to a non-parametric one with DRF. Meanwhile, we employ the distance metrics in the functional space of Partial DRF as the splitting criterion and use kernel-based DML estimators as the DRF approximation.

When it comes to implementations, many open source packages integrate recent research with the trending ML methods that facilitate HTE estimation, such as EconML (Oprescu et al.), CausalML (Chen et al., 2020) and GRF. EconML, CausalML offer a suite of high-performance ML algorithms like DML, while GRF specially supports CF. However, they are only compatible with single-machine systems and may not be efficient when it comes to large-scale instances. Apache Spark (Meng et al., 2016) is integrated for handling large-scale data with distributed and in-memory computing, while inclusively provides APIs of trending ML algorithms with the built-in MLlib. Therefore, as one of our contributions, we build our algorithm on Spark that enables large-scale data processing and easy use of any ML techniques. To summarize, the main contributions are:

- We are the first to generalize CF's partially linear model to a non-parametric one with DRF and apply kernel-based DML estimators. The proposed algorithm pictures the non-parametric behaviors of how outcome varies with treatments.

- We propose a distance-based splitting criterion with various distance metrics in the functional space of Partial DRF that generalizes the differences of CAPE in CF.

- Our proposed GCF has been tested on both synthetic and large-scale real-world datasets. It improves on SOTA in terms of multiple evaluation metrics. We implement GCF on Apache Spark MLlib and achieves higher computatinal efficiency by distributed computing.

The rest of the paper is organized as follows. In Section 2, we present related works in the literature. In Section 3, we introduce some notations and backgrounds. In Section 4, we introduce the Generalized Causal Forest. In Section 5, we examine the empirical performance of GCF by applying it to both synthetic and real-world data sets. We conclude the paper with some discussions in Section 6.

## 2 RELATED WORKS

A growing amount of literature has been devoted to address the problem of HTE estimation with continuous treatments. The algorithms in the context of DRF include methods targeting at confounding bias, kernel-based or ML-based methods for regression bias reduction, and techniques of DRE or DML that balance the trade-off between the two biases. DRF combined with IPW (Zhu et al., 2015; Graham et al., 2015) can achieve consistent estimators by weighting the estimation with probability density of treatment. For the regression bias, (Galagate & Schafer, 2015) employ parametric estimations and (Flores et al., 2007) model DRF as non-parametric functions using kernel regression. To advance, (Kennedy et al., 2017) propose a doubly robust estimators for DRF by combine the estimation of generalized propensity score with the estimation of outcome using kernel regression. Thus far, these approaches only provide global estimations and cannot handle massive amounts of high-dimensional data.

Recently, the great efficiency of ML methods motivates their generalization to the problem of HTE estimation. Towards that end, estimating DRF with ML-based algorithms is developed and among which tree-based models are a great candidate since it partitions the feature space for dimensionality reduction and maximizes the heterogeneity as well. Causal Forest (CF) proposed in (Athey et al., 2019) utilizes a subset of training samples for growing trees by recursively partitioning via a splitting criterion. Then HTE estimation is given by weighted average over the outcomes of the remaining training samples, known as honesty principle. The final estimator obtained from CF further exhibits a lower degree of bias by tree ensembles. However, their splitting criterion relies on a linear model assumption and is formulated as the difference between the slopes of linear models. The same limitation applies to Orthogonal Random Forest (Oprescu et al., 2019) and Bayesian Additive Regression Tress (Hill, 2011; Woody et al., 2020; Hahn et al., 2020). Nevertheless, the complexity of HTE estimation may not be fully captured by linear or general parametric models, which brings the necessity of non-parametric ML models.

Kernel-based DML (Colangelo & Lee, 2020) estimates the nuisance functions with cross-fitting, constructs a non-parametric DML estimator by Gaeutax Derivative. This motivates the utilization of DML and non-parametric estimation in our work, though it only provide global estimators with the limited capacity of localization. A fully non-parametric DML with locally weighted estimations for continuous treatments has not been considered yet.

Building on prior art, we point out that our integration of DRF estimation into CF overcome the challenges aforementioned by using local non-parametric DRF to constructing splitting criterion. We introduce the conceptual partial DRF as a component of the splitting criterion and employ the distance in the functional space of Partial DRF as a proxy for heterogeneity instead of the difference of slopes. Moreover, the partial DRF can be estimated precisely by the robust kernel-based DML estimators in our splitting criterion.

## 3 PREREQUISITES

### 3.1 NOTATIONS AND ASSUMPTIONS

We formally introduce the notations for HTE estimation with continuous treatments. Following the potential outcome framework in (Neyman, 1923; Rubin, 1974), we let $T$ be the continuous treatment, $\boldsymbol{X} = (X^j)_{j=1}^{p_X}$ be the $p_X$-dim confounding variables, $\boldsymbol{U}$ be the outcome-specific covariates, $\boldsymbol{Z}$ be the treatment-specific covariates that are independent of $\boldsymbol{U}$, and $Y$ be the outcome of interests. The population $\Omega = (\boldsymbol{X}, \boldsymbol{U}, \boldsymbol{Z}, Y, T) \in \mathbb{R}^{p_X + p_U + p_Z + 1 + 1}$ satisfies

$$Y = g(T, \boldsymbol{X}, \boldsymbol{U}) + \epsilon; T = f(\boldsymbol{X}, \boldsymbol{Z}) + \nu$$

where $\epsilon, \nu$ are noises of zero mean and $g : \mathbb{R}^p \times \mathbb{R} \to \mathbb{R}$ and $f : \mathbb{R}^p \to \mathbb{R}$.

$\{(\boldsymbol{X}_i, \boldsymbol{U}_i, \boldsymbol{Z}_i, Y_i, T_i), i = 1, \ldots, n\}$ are i.i.d. samples drawn from the population $\Omega$. In practice, the decomposition of $\boldsymbol{X}, \boldsymbol{U}, \boldsymbol{Z}$ from observed covariates is difficult. Therefore, without additional specification, we use $\boldsymbol{X}$ to represent observed covariates throughout the paper to simply notations. The potential outcomes under treatment $t$ is $Y_{(t)}$. Recall that Propensity Score (PS) (Rubin, 1974) for a discrete treatment is $P(T = t | \boldsymbol{X})$, the probability for a unit receiving treatment $t$ given the covariates $\boldsymbol{X}$. For continuous treatments, (Rubin, 1974; Hirano & Imbens, 2004) introduce Generalized Propensity Score (GPS) which is the probability density function $\pi(T = t | \boldsymbol{X})$.

The estimand of interests, CATE $\theta(t, \boldsymbol{X})$, is formally defined as

$$\theta(t, \boldsymbol{X}) = E[Y_{(t)}|\boldsymbol{X}] - E[Y_{(0)}|\boldsymbol{X}]$$

To identify $\theta(t, \boldsymbol{X})$ , common assumptions as in (Holland, 1986; Kennedy et al., 2017) are made throughout the paper.

**Assumption 1.** *Consistency: $E[Y|T = t] = E[Y_{(t)}|T = t]$, i.e. the outcome of any sample solely depends on its treatment;*

**Assumption 2.** *Ignorability: The potential outcomes $Y_{(T)}$ is independent of treatment $T$ given covariates $\boldsymbol{X}$.*

**Assumption 3.** *Positivity: The GPS $\pi(T = t|\boldsymbol{X}) > p_{min} > 0$, $\forall t, \boldsymbol{X}$, i.e. the density for any sample receiving any treatment is bounded away from 0.*

Under the above assumptions, we have

$$\theta(t, \boldsymbol{X}) = E[Y_{(t)}|\boldsymbol{X}] - E[Y_{(0)}|\boldsymbol{X}] = E[Y|T = t, \boldsymbol{X}] - E[Y|T = 0, \boldsymbol{X}]$$
$$= E[g(t, \boldsymbol{X})|T = t, \boldsymbol{X}] - E[g(t, \boldsymbol{X})|T = 0, \boldsymbol{X}]$$

where the first equality holds by Assumption 1 and the second equality holds by Assumption 2. Positivity is indispensable for the conditional expectation in the last line to be well-defined whereas being often too strong. In practice, it can be reduced to the following.

**Assumption 4.** *Weak Positivity: The variance of GPS $\sigma(\pi) = \int_t t^2 \cdot \pi(T = t|\boldsymbol{X})dt - \left(\int_t t \cdot \pi(T = t|\boldsymbol{X})dt\right)^2 > \sigma_{min} > 0$, $\forall t, \boldsymbol{X}$.*

## 3.2 DOSE-RESPONSE FUNCTION

For continuous treatments, the treatment effect can be characterized by DRF, formally defined as

$$\mu(t) := E[Y|T = t] \tag{1}$$

Generally speaking, DRF can be either parametric or non-parametric, with parametric ones being applied to CATE estimation as in (Robins, 2000; Van der Laan et al., 2003). Non-parametric ones can better explore the functional space of ADRF out of flexibility and hence being the tool of our proposed algorithm.

The target estimand of DRF heavily depends on the contexts (Galagate, 2016), such as treatment effect $\left(E[Y_{(t)}] - E[Y_{(0)}]\right)$, partial effect $\left(\frac{E[Y_{(t_2)}] - E[Y_{(t_1)}]}{t_2 - t_1}\right)$ and the elasticity $\left(\frac{\partial log(E[Y_{(t)}])}{\partial log(t)}\right)$. Specifically, conditional DRF (CDRF) $\mu(t, \boldsymbol{X}) = E[Y|T = t, \boldsymbol{X}]$ characterizes the treatment effect at individual levels and thereby leading to a CATE estimator as $\theta(t, \boldsymbol{X}) = \mu(t, \boldsymbol{X}) - \mu(0, \boldsymbol{X})$, which is the target estimand hereafter.

## 3.3 KERNEL REGRESSION AND DOUBLE/DEBIASED MACHINE LEARNING

For non-parametric modeling, kernel regression (Fan & Gijbels, 2018) works with solid theoretical foundations. Specifically, given a non-parametric model $y = g(\boldsymbol{x}) + \epsilon$ with data $(\boldsymbol{X}_i, Y_i)$, kernel regression with linear kernel smoothers gives the estimator $\hat{g}(\boldsymbol{x}) = \frac{\sum_{i=1}^{n} K_h(\boldsymbol{x}, \boldsymbol{X}_i) \cdot y_i}{\sum_{i=1}^{n} K_h(\boldsymbol{x}, \boldsymbol{X}_i)}$ where $K_h(\boldsymbol{x}, \boldsymbol{X}_i)$ is a scaled kernel density with $h$ being the bandwidth for smoothness. Typical choices of kernel include Uniform, Epanechnikov, Biweight, Triweight and Gaussian.

The kernel-based DML estimator proposed in (Colangelo & Lee, 2020) performs a 2-stage DRF estimation maintaining asymptotically normal. The DML estimator for the DRF $\mu(t)$ is proposed as follows. First, we estimate the CDRF $\mu(t, \boldsymbol{X})$ with $\hat{\mu}(t, \boldsymbol{X})$ and the GPS $\pi(T = t|\boldsymbol{X})$ with $\hat{\pi}(t|\boldsymbol{X})$. Then, with the nuisance functions plugged in, kernel-based double/debiased estimator gives

$$\hat{\mu}(t) = \frac{1}{n} \cdot \sum_{i=1}^{n} \hat{\mu}(t, \boldsymbol{X}_i) + \frac{K_h(T_i - t)}{\hat{\pi}(t|\boldsymbol{X}_i)} \cdot (Y_i - \hat{\mu}(t, \boldsymbol{X}_i))$$

# 4 GENERALIZED CAUSAL FOREST

In this section, we formally present the details of our algorithm, GCF. Building on CF, GCF relaxes partially linear assumption by adapting non-parametric DRF to the splitting criterion that can be estimated using kernel-based DML estimator described in the previous section. In what follows, we give it to a workflow of GCF at both the training stage and prediction stage, followed by the elaborations on the details of splitting criterion, estimations, and practical tweaks for implementation.

---

**Algorithm 1:** Generalized Causal Forest

---

**Input** Data set $O = (\boldsymbol{X}_i, T_i, Y_i), i = 1, \ldots, n$; number of trees $B$; number of features sampled
 for growing a tree $mtry$; the minimum sample size on each leaf node $min.node.size$; honesty
 fraction $\alpha$; tolerance $\tau$; positivity threshold $\zeta$;
**Trainings begin**
> Data $O_1$ and data $O_2$ by a $(\alpha, 1-\alpha)$ split rule applied to $O$ for honesty;
> Pre-train outcome regression model $\hat{\mu}$ and treatment density estimation $\hat{\pi}$ on sample $\Omega_1$;
> $b \longleftarrow 1$;
> **while** $b \leq B$ **do**
>> Randomly sample a feature set of size $mtry$ $\boldsymbol{X}^S$ from $\boldsymbol{X}$;
>> **for** *The stopping rule is not satisfied* **do**
>>> Identify each parent node $P$;
>>> Compute the splitting criterion $\Delta(\cdot)$ (3) over the samples in $O_1$ with $\boldsymbol{X}^S$;
>>> Grow the tree $\mathcal{T}_b$ by splitting at the parent node $P$ according to the $\Delta(\cdot)$;
>> **end**
>> Assign samples in $O_2$ to leaf nodes based on $\mathcal{T}_b$ and let $b \longleftarrow b+1$;
> **end**
**end**
**Output** Causal forest with $B$ trees by recursive partitioning on $X$; node assignments for
 samples in $O_2$;
**Predictions begin**
> CDRF estimation $\hat{\mu}_b(t, \boldsymbol{x})$ by local weighted average over the outcomes of samples in $\Omega_2$
> that falls into $\mathcal{L}_b(\boldsymbol{x})$ as $\sum_{i=1}^n \frac{1_{X_i \in \mathcal{L}_b(\boldsymbol{x})} \cdot 1_{T_i=t} \cdot Y_i}{|\mathcal{L}_b(\boldsymbol{x})|}$; CATE estimation
> $\hat{\theta}_b(t, \boldsymbol{x}) = \hat{\mu}_b(t, \boldsymbol{x}) - \hat{\mu}_b(0, \boldsymbol{x})$ and $\hat{\theta}(t, \boldsymbol{x}) = \frac{1}{B} \sum_{b=1}^B \hat{\theta}_b(t, \boldsymbol{x})$;
**end**

---

We introduce the partial derivative of DRF, PDRF at treatment $t$ with a slightly abuse use of notations

$$\Phi(t) := \left. \frac{\partial \mu(T)}{\partial T} \right|_{T=t} \quad \text{and} \quad \Phi(t, \boldsymbol{X}) := \left. \frac{\partial \mu(T, \boldsymbol{X})}{\partial T} \right|_{T=t}.$$

Intuitively, it tells about how $\mu(\cdot)$ varies with treatments and characterizes the "slope" along $t$. Different leaf nodes on a tree shall have different slopes as the heterogeneity of covariates $\boldsymbol{X}$.

Let the number of trees be $B$ and then GCF grows trees $\mathcal{T}_b, b = 1, \ldots, B$, by repeating tree-growing process $B$ times with bootstrapping. At the training stage, with PDRF we construct the tree $\mathcal{T}_b$ by recursive partitioning based on a splitting criterion $\Delta(c_1, c_2)$. It is proportional to the discrepancy between the PDRF in the left and right child nodes, to maximize the heterogeneity. We elaborate on this splitting criterion in the next part.

Our algorithm is implemented on Apache Spark for large-scale data processing and the mechanism of the tree-growing process is different from that in CF. To elucidate on that, the data is stored at the master machine and trees are copied to each branch machine. Then data is randomly distributed to branch machines for the computation and collected back to the master machine for the final criterion that is used for updating a tree. The updated tree will be copied to each branch machine again and we repeat the process until the tree stops splitting. This distributed computation and clustered storage leverage the computational power of multiple machines for speeding up the training process. We refer readers to the appendix for more information on GCF's Spark implementation.

The splits or the tree-growing process terminate when various pre-specified stopping criteria are met, for example, the sample size of child nodes is smaller than the parameter $min.node.size$ or the distance $D$ between PDRF is smaller than $min.info.gain$ $\tau$.

For prediction, we also follow the honesty principle proposed in (Athey et al., 2019) by partitioning the training samples into two parts where one is for constructing the tree and the other is for estimating CATE on the leaf node. That is to say that each training data can either be utilized to estimate CATE or contribute to growing a tree. At the prediction stage, the leaf node of given samples $\boldsymbol{x}$ on each tree $\mathcal{T}_b$ is denoted by $\mathcal{L}_b(\boldsymbol{x})$. The estimation of CDRF $\hat{\mu}_b(t, \boldsymbol{x})$ on $\mathcal{L}_b(\boldsymbol{x})$ takes a local weighted average on training samples in $\mathcal{L}_b(\boldsymbol{x})$, which is given by $\sum_{i=1}^n \frac{1_{X_i \in \mathcal{L}_b(\boldsymbol{x})} \cdot 1_{T_i=t} \cdot Y_i}{|\mathcal{L}_b(\boldsymbol{x})|}$ and $\hat{\theta}_b(t, \boldsymbol{x}) = \hat{\mu}_b(t, \boldsymbol{x}) - \hat{\mu}_b(0, \boldsymbol{x})$. The final CATE estimation for $X$ is the average of $\mathcal{L}_b(\boldsymbol{x})$ over $B$ trees, as $\hat{\theta}(t, \boldsymbol{x}) = \frac{1}{B} \sum_{b=1}^B \hat{\theta}_b(t, \boldsymbol{x})$.

### 4.1 SPLITTING CRITERION

Given a parent node $P$ and training samples with covariates $\boldsymbol{X}_\omega$, our splitting criterion $\Delta(c_1, c_2)$ is proposed as

$$\Delta(c_1, c_2) = \frac{n_{c_1} n_{c_2}}{n_P^2} D(\hat{\Phi}_{c_1}(t, \boldsymbol{X}_\omega), \hat{\Phi}_{c_2}(t, \boldsymbol{X}_\omega)), \tag{2}$$

where $\hat{\Phi}_{c_1}$, $\hat{\Phi}_{c_2}$ are the PDRF estimators, respectively. The sample sizes of parent node, left child node and right child node are $n_P$, $n_{c_1}$ and $n_{c_2}$, respectively. The ratio $\frac{n_{c_1} n_{c_2}}{n_P^2}$ in $\Delta(c_1, c_2)$ is to balance the sample sizes between the two child nodes. Distance metric $D$ measures the distance between $\hat{\Phi}_{c_1}(t, \boldsymbol{X}_\omega)$ and $\hat{\Phi}_{c_2}(t, \boldsymbol{X}_\omega)$ in the functional space of PDRF, which is to be maximized in the splitting so as to optimize the heterogeneity. Some commonly used metrics are $L_1$, $L_2$ and $L_\infty$ as in (Dette et al., 2018) $D_1 = \int_t |\hat{\Phi}_{c_1}(t, \boldsymbol{X}_\omega) - \hat{\Phi}_{c_2}(t, \boldsymbol{X}_\omega)| dt$, $D_2 = \int_t |\hat{\Phi}_{c_1}(t, \boldsymbol{X}_\omega) - \hat{\Phi}_{c_2}(t, \boldsymbol{X}_\omega)|^2 dt$, $D_\infty = max_t |\hat{\Phi}_{c_1}(t, \boldsymbol{X}_\omega) - \hat{\Phi}_{c_2}(t, \boldsymbol{X}_\omega)|$. We next show how to estimate PDRF $\Phi_{c_1}$ and $\Phi_{c_2}$ using kernel-based DML estimator (Colangelo & Lee, 2020).

### 4.2 PDRF ESTIMATION

The estimation for DRF is a precondition for the estimation of PDRF. In this paper, we point out kernel-based DML estimator in (Colangelo & Lee, 2020) can be adapted to the DRF estimation in our GCF and guides the tree splitting.

Recall that, the DML estimation for DRF is given by

$$\hat{\mu}(t) = \sum_{i \in C} \left( \hat{\mu}(t, \boldsymbol{X}_i) + \frac{K_h(T_i - t)}{\hat{\pi}(t|\boldsymbol{X}_i)} \cdot (Y_i - \hat{\mu}(t, \boldsymbol{X}_i)) \right)$$

where $\hat{\mu}(t, \boldsymbol{X}_i)$ is the pretrained estimator for $\mu(t, \boldsymbol{X})$ at $\boldsymbol{X}_i$, $\hat{\pi}(t|\boldsymbol{X}_i)$ is the pretrained estimator for GPS at $\boldsymbol{X}_i$ and $C$ denotes the child node.

When adapted to our framework, we additionally derive the following estimator for CDRF by innovatively combine kernel regression with the DML estimator.

**CDRF estimator**  The estimator is given by

$$\tilde{\mu}(t, \boldsymbol{X}_i) = \hat{\mu}(t, \boldsymbol{X}_i) + \frac{K_h(T_i - t)}{\hat{\pi}(t|\boldsymbol{X}_i)} \cdot (Y_i - \hat{\mu}(t, \boldsymbol{X}_i))$$

**PDRF estimator**  The PDRF estimator is given by $\hat{\Phi}(t) = \frac{\partial \tilde{\mu}(t)}{\partial t}$ and $\hat{\Phi}(t, \boldsymbol{x}) = \frac{\partial \tilde{\mu}(t, \boldsymbol{x})}{\partial t}$.

In estimating PDRF, the smoothness condition of DRF plays an important role. For smooth DRF, we can explicitly derive the closed-form PDRF. For smooth Gaussian kernels, $\Phi(t)$ is

$$\hat{\Phi}(t) = \frac{\partial \tilde{\mu}(t)}{\partial t} = \frac{\sum_{i=1}^n K_h'(T_i - t) Y_i}{\sum_{i=1}^n K_h(T_i - t)} - \Big[ \sum_{i=1}^n K_h(T_i - t) Y_i \Big] \frac{\sum_{i=1}^n K_h'(T_i - t)}{[\sum_{i=1}^n K_h(T_i - t)]^2}$$

However, for general DRF, the PDRF may be discontinuous or even ill-defined. For example, non-differentiable kernel functions like uniform, epanechnikov, biweight and triweight, $\Phi$ can only be estimated via numerical approximations of $\Phi(t) = \dfrac{E[\mu(t + \delta) - \mu(t)]}{\delta}$.

### 4.3 PRACTICAL CONSIDERATIONS

In the work of (Kennedy et al., 2017), they claim that the choice of bandwidth $h$ for kernels weighs more than the choice of density functions, since $h$ deals with the trade off between bias and variance. Usually, a small $h$ avoids a large variance while a large $h$ reduces the bias. The empirical ways of choosing an optimal $h$ are Cross Validation and Rule-of-thumb (Dehnad, 1987)

When utilizing kernel densities for weighting, we often suffer from the boundary bias if without normalization. To this end, we normalize estimators with the cumulative density function of range $[T_{min}, T_{max}]$ of treatments. More specifically, the estimators are divided by $\int_{T_{min}}^{T_{max}} K_h(T_i - t)dt$.

Regarding the positivity assumption in practice, we use a hyperparameter $\zeta$ to regularize for the boundary of $\sigma(\pi)$ that is the variance of GPS. This is also a guarantee on the weak assumption as stated in Assumption 4. Formally, our proposed splitting criterion will be a regularized one as

$$\Delta(c_1, c_2) = \frac{n_{c_1} n_{c_2}}{n_P^2} D(\hat{\Phi}_{c_1}(t, \boldsymbol{X}_\omega), \hat{\Phi}_{c_2}(t, \boldsymbol{X}_\omega)) + \zeta \cdot \sigma(\hat{\pi}), \tag{3}$$

where $\hat{\pi}$ is the estimated GPS. With this additional term, the splitting will tend to maximize the variance of GPS and hence taking the variety of treatments across samples into account.

### 4.4 ASYMPTOTIC PROPERTY

Let Assumptions 1-3 in (Colangelo & Lee, 2020) hold and thusfar the convergence of kernel-based DML estimator is naturally guaranteed at a rate of $\sqrt{n}$ as long as the outcome is unbiased or the generalized propensity score is unbiased. That is to say that we can utilize a subsample $s$ of order $\sqrt{n}$ to achieve a converged kernel DML estimator. Then $s$ satisfies that $s/n \rightarrow 0$ and $s \rightarrow \infty$ which is stated as Specification 1 in (Athey et al., 2019). Under the above assumptions and suppose one additional condition be that the score function is lipschitz with a bounded support of treatment, then Assumption 1-6 in (Athey et al., 2019) automatically hold for the score function. Throughout our algorithm, we also follow the honesty principle. Then by Theorem 3 (Athey et al., 2019), our final estimator is asymptotically unbiased. This shows the doubly robustness of our estimator by combining kernel DML and Causal Forest under slightly stronger assumptions on the score function.

## 5 EXPERIMENT

In this section, we validate GCF by showing that 1) GCF can provide CATE estimations in a variety of contexts such as simulated data with true labels and real-world data without the ground truth 2) GCF is flexible yet effective compared to SOTA which provides enough evidence for its practical use. More specifically, we present the detailed experimental results of our proposed algorithm and demonstrate its effectiveness compared to the prior art on synthetic data and real-world data sets.

### 5.1 EVALUATION

In what follows we introduce the commonly used metrics for model evaluation on both synthetic data and real-world data sets, named as model errors and Qini score and Qini curve, respectively. For real-world data sets, the model errors are not available since the counterfactual outcomes in the potential outcomes cannot be observed.

**Synthetic Data**   We adopt PEHE and RMSE as the measure for model precision. Formally, let $n$ be the number of treatment groups, $\hat{\theta}_i^t$ and $\theta_i^t$ be the predicted and true treatment effect of $i_{th}$ sample at treatment $t$, $\hat{\theta}^t = \sum_{i=1}^t \hat{y}_i^t - \hat{y}_i^0$, $\hat{y}_i^0$ be the reference outcome that is pre-determined.

PEHE,RMSE,and ADRF are $\dfrac{\sum_{i=1}^n (|\hat{\theta}^t - \theta^t|))}{n}$, $\sqrt{\dfrac{\sum_{t=1}^n (\hat{\theta}^t - \theta^t)^2}{n}}$, $\hat{y(t)} = \int_x \hat{\mu}(t, x)dx$.

**Real-world data sets** We refer readers to (Gutierrez & Gérardy, 2017) for the formal definition of Qini Score and Qini Curve. In practice, a larger Qini Score indicates that model has better performance in identifying HTE.

## 5.2 MODEL SETUPS

The baselines for GCF include RF, CF, and Kennedy (Kennedy et al., 2017). Both RF and CF use $num.trees$ equals to 500, $min.node.size$ to 50. For Kennedy (Kennedy et al., 2017), we use the code of function $cts.eff$ from R package $npcausal$ available in (ehkennedy, 2017). SuperLearner library is set to $SL.ranger$ and $SL.glm$. We made small tweaks to $SL.ranger$ to make share same hyperparaters with RF and CF for comparability.

**GCF** We implement the proposed GCF on Apache Spark MLlib (Meng et al., 2016) with kernel-based DML estimator, denoted by GCF. For the implementation of GCF, we fix the distance metric $D$ to be $d_2$ out of simplicity. The number of trees, the minimum node size and other hyperparameters are set to match those in RF and CF.

RF is a forest-ensemble method without controlling confoundedness. CF makes the partially linear assumption on DRF. Kennedy is a non-parametric doubly robust algorithm but uses a normal splitting criterion. The comparisons inclusively validate our algorithm for both non-parametric DRF estimation and specially designed splitting criterion.

## 5.3 SIMULATION

Here we show a comprehensive simulation study on synthetic data. First, we elaborate on the data generating process. It is followed by the details of model implementations. Finally we illustrate the numerical results that consistently show the superior performance of our GCF.

### 5.3.1 DATA GENERATING PROCESS

Let $n$ denote the number of samples and $p = p_X + p_U + p_Z$ be the dimension of covariates. The covariate matrix is $(X_i^j)_{i=1,...,n}^{j=1,...,p_X} = (\boldsymbol{X}_1, \ldots, \boldsymbol{X}_n) \in \mathbb{R}^{n \times p_X}$. Data generating process (DGP) is

$$Y = \mu(T) + 0.2(X^1 \cdot X^1 + X^4)T + \boldsymbol{X} \cdot \boldsymbol{\beta_X} + \boldsymbol{U} \cdot \boldsymbol{\beta_U} + \epsilon$$
$$T = 20 \cdot \Psi(\phi(\boldsymbol{X} \cdot \boldsymbol{\beta_X} + \boldsymbol{Z} \cdot \boldsymbol{\beta_Z})) + \nu$$

where $\phi$ is the sigmoid function and $\Psi$ is the pdf of Beta distribution. Here we set DRF $\mu$ to be polynomial(Poly), exponential(Exp), and sinusoidal functions(Sinus).

The above DGP gives us a total of 6 datasets by taking the possible combinations of 3 DRFs $\mu(t)$ and 2 setups of covariate $\Sigma$. Noises $\epsilon$, $\nu$ follow $Unif(-1, 1)$. Covariates $\boldsymbol{X}$ and coefficients $\beta$ follow $N(0, I_{p_X})$. We also allow sparsity in covariates by randomly setting some of the coefficients to 0. Testing data sets are generated with randomly assigned treatments to allow unbiased evaluation.

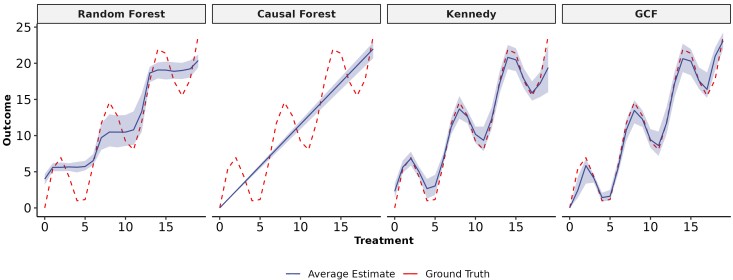

Figure 2: Comparison of ADRF estimates for RF, CF, Kennedy and GCF. DRF follows sinusoidal pattern. Blue line denotes model average estimations. Confidence bounds of model estimates generated by multiple simulation run are marked in light-blue shade. Ground truth of testing data set is represented in red dash-line.

### 5.3.2 SIMULATION RESULTS

Since some models output the final prediction of $\hat{Y(t)}$ and some output the treatment effect $\hat{\theta(t)}$, we deliberately compute the pseudo outcome $\tilde{Y}(t) = Y(t) - Y(0)$ as the label in the purpose of the comparability across baselines. The simulation results pf 3 datasets are summarized in Table 1 and those of the other 3 datasets are in Appendix. Here PEHE and RMSE are averaged and standard deviations of the metrics across simulation runs are also reported. Overall, GCF significantly outperforms baseline methods, which again validates the superiority of the newly proposed algorithm. More specifically, GCF exhibits the smallest biases across multiple DGP setups which we defer details to the supplementary.

Additionally, we compare the ADRF curve given by different models to the ground truth. An example of ADRF under a DGP specification is shown in Fig. 2.

Table 1: Simulation results on datasets with different DRFs where number of samples $n = 1000, p_X = 50, p_Y = 5, p_Z = 5$. Standard deviations are in parenthesis over 100 simulations.

|  | *Polynomial* | | *Sinusoidal* | | *Exponential* | |
|---|---|---|---|---|---|---|
| **Methods** | **PEHE** | **RMSE** | **PEHE** | **RMSE** | **PEHE** | **RMSE** |
| RF | 5.63(0.4) | 4.61(0.3) | 4.27(0.4) | 3.21(0.2) | 3.57(0.4) | 2.62(0.2) |
| CF | 14.09(0.4) | 12.58(0.4) | 5.15(0.4) | 3.96(0.2) | 4.34(0.3) | 3.37(0.2) |
| Kennedy | 4.37(0.5) | 3.36(0.5) | 4.14(0.5) | 2.78(0.3) | 3.86(0.4) | 2.54(0.2) |
| **GCF** | **4.14(0.3)** | **2.88(0.2)** | **4.05(0.4)** | **2.7(0.3)** | **3.85(0.4)** | **2.48(0.2)** |

### 5.4 REAL-WORLD DATA SETS

Our method is also implemented on real-world data sets with size of 10,698,884 which sourced from a leading ride-hailing company. A discount that randomly sampled (one of 6 different values $\{d_0, d_1, d_2, d_3, d_4, d_5 : d_0 < d_1 < \ldots < d_5\}$) is assigned to all trips in a given origin-destination-time tuple. The effect of discount on the trip complete rate is of primary interests. We apply GCF to this data set and evaluate the performance of models by calculating Qini Scores.

Table 2: Qini scores of models under different treatments

| Methods | $d_5$ | $d_4$ | $d_3$ | $d_2$ | $d_1$ |
|---|---|---|---|---|---|
| Xgboost | 0.253 | 0.171 | 0.177 | 0.206 | 0.177 |
| CF | 0.253 | 0.194 | 0.202 | 0.272 | 0.300 |
| **GCF** | **0.309** | **0.248** | **0.305** | **0.444** | **0.780** |

The Qini scores of different models are summarized in Table 2. The performance of GCF is superior than the company's current models as GCF has the highest Qini scores across all levels of discounts. We further create an incentive policy based on GCF output and deploy it for online purpose. Online A/B testing shows a significant lift in KPI for GCF-driven policy.

## 6 CONCLUSION

In this paper, we propose a novel forest-based non-parametric algorithm to address the problem of HTE estimation with continuous treatments. Under the situation that a fully non-parametric and localized ML-based algorithm has not been proposed yet, we extend Causal Forest with a DRF-based splitting criterion computed by the distance in the functional space of PDRF. To estimate this DRF, We use the kernel-based DML estimator to guarantee double robustness that mitigates model misspecifications. We implement GCF on Spark to leverage the computational efficiency and test it on both synthetic data and real-world data sets and compare it with the prior art approaches. The numerical results demonstrate that our method significantly outperform competing methods.

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

## 7  APPENDIX

The appendix runs as follows. First, we list all the notations used in the appendix. Then we dive into the details of the data generating process for the simulation study and present additional simulation results which further show the effectiveness of our proposed method. Subsequently, we briefly outline GCF's Spark framework, which enables large-scale instances processing and faster model training. Finally, we demonstrate how to apply our method to an online schema and achieve superior performance as its another validation.

### 7.1  NOTATIONS

We restate the notations that are consistent with the main paper.

Following the potential outcome framework in (Neyman, 1923; Rubin, 1974), we let $T$ be the continuous treatments, $\boldsymbol{X} = (\boldsymbol{X}^j)_{j=1}^p$ be the $p_X$-dim confounder, $\boldsymbol{U}$ be the $p_U$-dim outcome-specific adjustment variable, $\boldsymbol{Z}$ be the $p_Z$-dim treatment-specific adjustment variable, and $Y$ be the observed outcome. The potential outcomes under treatment $t$ is $Y_{(t)}$. The population $\Omega = (\boldsymbol{\Sigma} = (\boldsymbol{X}, \boldsymbol{U}, \boldsymbol{Z}), Y, T) \in \mathbb{R}^{p_X + p_U + p_Z + 1}$ satisfies

$$Y = g(T, \boldsymbol{X}, \boldsymbol{U}) + \epsilon$$
$$T = f(\boldsymbol{X}, \boldsymbol{Z}) + \nu$$

where $\epsilon, \nu$ are noises of zero mean,

and $g : \mathbb{R}^{p_X + p_U} \times \mathbb{R} \to \mathbb{R}$ and $f : \mathbb{R}^{p_X + p_Z} \to \mathbb{R}$.

$\{(\boldsymbol{\Sigma}_i, Y_i, T_i), i = 1, \ldots, n\}$ are i.i.d. samples drawn from the population $\Omega$. Then the covariate matrix $\Sigma = (\Sigma_i^j)_{1 \le i \le n}^{1 \le j \le p} = (\boldsymbol{\Sigma}_1, \ldots, \boldsymbol{\Sigma}_n) \in \mathbb{R}^{n \times p}$ where $p = p_X + p_U + p_Z$. The Generalized Propensity Score is $\pi(T = t | \boldsymbol{X})$, which is the probability density for a unit receiving treatment $t$ given the covariate $\boldsymbol{X}$.

### 7.2  DATA GENERATING PROCESS

Recall that the covariate matrix $\boldsymbol{\Sigma} = (X_i^j, U_i^j, Z_i^j) = (\boldsymbol{X}_1, \ldots, \boldsymbol{X}_n, \boldsymbol{U}_1, \ldots, \boldsymbol{U}_n, \boldsymbol{Z}_1, \ldots, \boldsymbol{Z}_n) \in \mathbb{R}^{n \times (p_X + p_U + p_Z)}$ and treatment $T \in \mathbb{R}$ where $n$ denotes the number of observations and $p$ is the dimension of covariates. Data generating process (DGP) works as

$$Y = \mu(T) + 0.2(\boldsymbol{X_1^2} + \boldsymbol{X_4})T + \boldsymbol{X} \cdot \boldsymbol{\beta_X} + \boldsymbol{U} \cdot \boldsymbol{\beta_U} + \epsilon$$
$$T = 20 \cdot \Psi(\phi(\boldsymbol{X} \cdot \boldsymbol{\beta_X^*} + \boldsymbol{Z} \cdot \boldsymbol{\beta_Z})) + \nu$$

where $\phi$ is the sigmoid function and $\Psi$ is the pdf of Beta distribution with shape parameters set to 2 and 3. Here we set DRF $\mu$ as polinomial(Poly), exponential(Exp), and sinusoidal functions(Sinus).

$$\mu(t) = 0.2 * (t - 5)^2 - t - 5, \text{ Polynomial}$$
$$\mu(t) = log(1 + exp(t)/(t + 0.1)) - log(11), \text{ Exponential}$$
$$\mu(t) = 5 * sin(x) + x, \text{ Sinusoidal}$$

The above DGP gives us a variety of data sets by taking the combination of DRF $\mu(t)$ and covariate $\Sigma$ that are specified by $n, p$ and $DRF$. Parameter $\epsilon$, $\nu$ are noises following $N(0, 1)$. Covariates $\boldsymbol{X}, \boldsymbol{U}, \boldsymbol{Z}$ follows $N(0, \boldsymbol{I})$ and coefficient vector $\boldsymbol{\beta}$ follow $[Unif(-1, 1)]$. Following this rule, we generate 100 rounds. Meanwhile, in test data, we randomly assigned the treatments to make sure unbiased evaluations.

### 7.3  SIMULATION RESULTS

In addition to the results provided in the main paper, here we provide the inclusive results of our simulation study, which covers all 6 data sets from the Data Generating process.

The results are summarized in Table 3. Compared to CF (Athey et al., 2019) that is the benchmark of our generalization, our GCF outperforms it with significantly smaller errors. With the baseine fixed to be Kennedy (Kennedy et al., 2017), our algorithm almost enjoys smaller errors except one dataset

Table 3: Simulation Results on Different data sets

*Setup: 1k,100,100,10,10*

| | Polynomial | | Sinusoidal | | Exponential | |
|---|---|---|---|---|---|---|
| **Methods** | **PEHE** | **RMSE** | **PEHE** | **RMSE** | **PEHE** | **RMSE** |
| RF | 8.08(0.4) | 7(0.4) | 4.89(0.4) | 3.74(0.2) | 4.28(0.4) | 3.3(0.2) |
| CF | 13.58(0.4) | 12.04(0.4) | 5.13(0.3) | 3.96(0.2) | 4.27(0.3) | 3.25(0.2) |
| Kennedy | 4.36(0.4) | 3.27(0.3) | 4.16(0.4) | 2.75(0.2) | 3.9(0.4) | 2.56(0.2) |
| **GCF** | **4.3(0.3)** | **3.01(0.2)** | **4.19(0.4)** | **2.86(0.2)** | **3.91(0.4)** | **2.53(0.2)** |

*Setup: 1k,100,50,5,5*

| | Polynomial | | Sinusoidal | | Exponential | |
|---|---|---|---|---|---|---|
| **Methods** | **PEHE** | **RMSE** | **PEHE** | **RMSE** | **PEHE** | **RMSE** |
| RF | 5.63(0.4) | 4.61(0.3) | 4.27(0.4) | 3.21(0.2) | 3.57(0.4) | 2.62(0.2) |
| CF | 14.09(0.4) | 12.58(0.4) | 5.15(0.4) | 3.96(0.2) | 4.34(0.3) | 3.37(0.2) |
| Kennedy | 4.37(0.5) | 3.36(0.5) | 4.14(0.5) | 2.78(0.3) | 3.86(0.4) | 2.54(0.2) |
| **GCF** | **4.14(0.3)** | **2.88(0.2)** | **4.05(0.4)** | **2.7(0.3)** | **3.85(0.4)** | **2.48(0.2)** |

where DRF is sin. Overall, these experimental evidence demonstrates that our GCF is superior than the algorithms for continuous treatments.

Meanwhile, ADRF curves are also inclusively presented here as validations of our proposed GCF. Figure 3−8 are ADRF curves of baseline methods and GCF on different synthetic data sets. Across all the data sets, our proposed GCF performs the best by being the closest to the ground truth.

## 7.4 SPARK IMPLEMENTATION

Apache Spark (Meng et al., 2016) has the power of large-scale data processing and provides APIs of any machine learning algorithms. Consequently, we build our proposed algorithm on Spark to do parallel computing and distributed model training that leverage the resources of multiple machines simultaneously.

The workflow of Spark with GCF is shown in Figure 9, which include integrated data and distributed computation. More specifically, with the parallel structure of spark, the tree-growing process runs and the process of GCF is depicted in Figure 10, which is significantly different from that in CF (Athey et al., 2019) by distributing tasks for computations

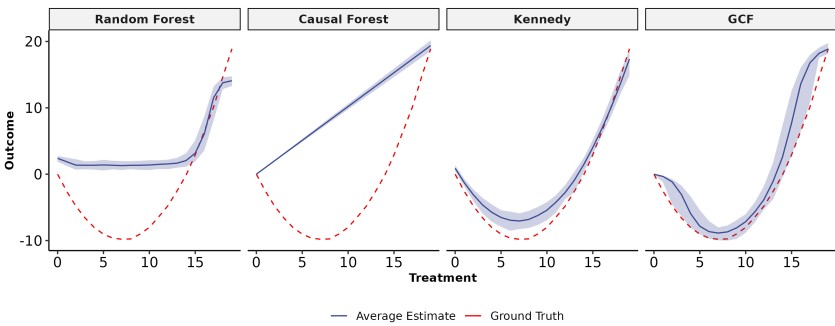

Figure 3: ADRF estimates . Blue line denotes model average estimations. Red dashline represents

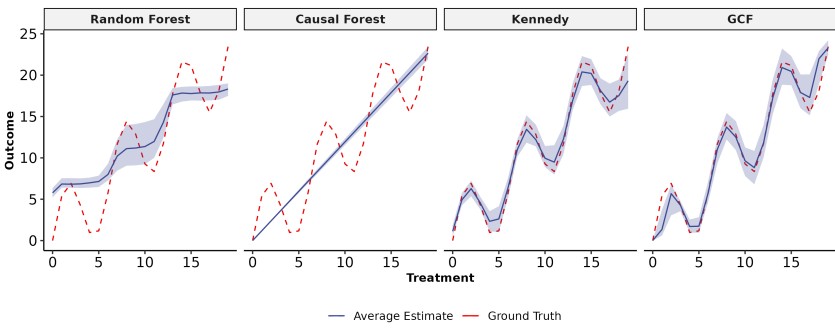

Figure 4: ADRF estimates . Blue line denotes model average estimations. Red dashline represents

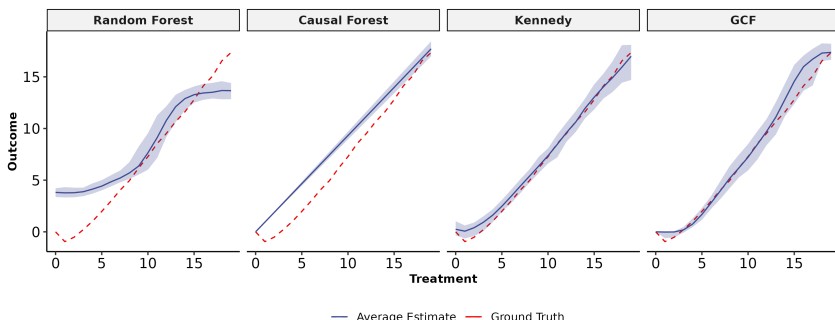

Figure 5: ADRF estimates . Blue line denotes model average estimations. Red dashline represents

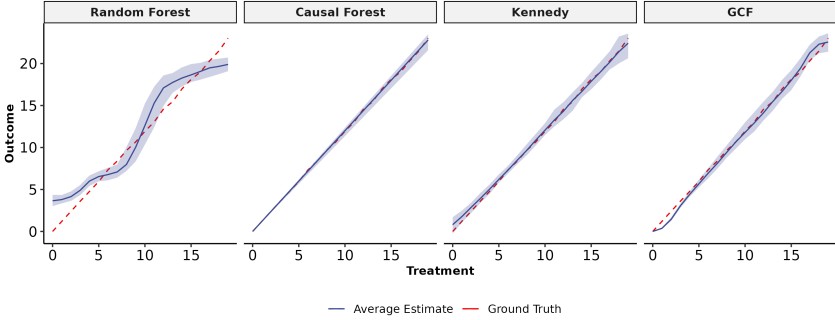

Figure 6: ADRF estimates . Blue line denotes model average estimations. Red dashline represents

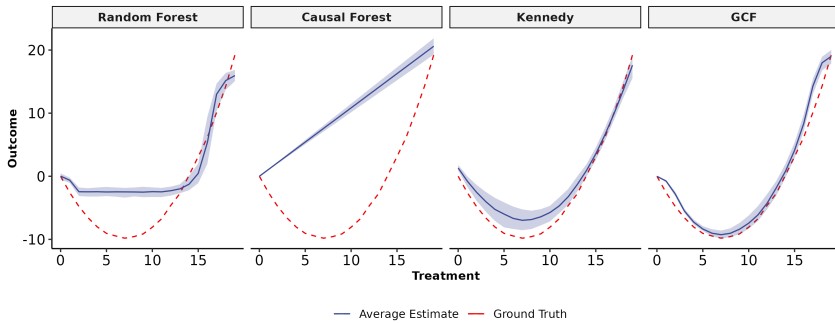

Figure 7: ADRF estimates . Blue line denotes model average estimations. Red dashline represents

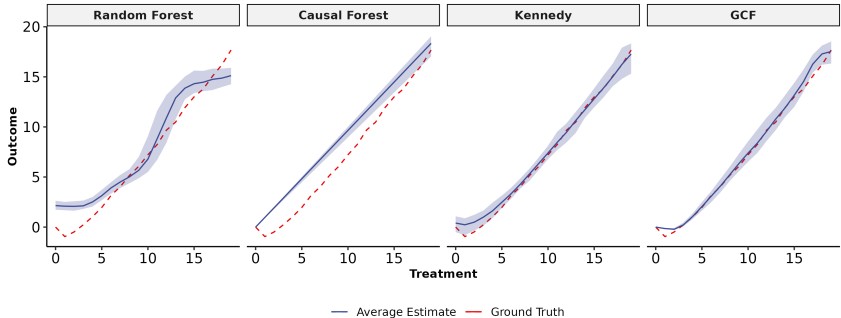

Figure 8: ADRF estimates . Blue line denotes model average estimations. Red dashline represents

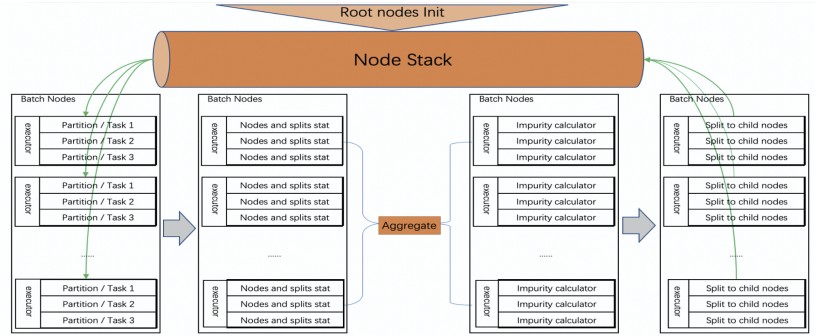

Figure 9: Parallel architecture of GCF Spark Implementation. Data are partitioned at the nodes level and distributed across executors to achieve high performance

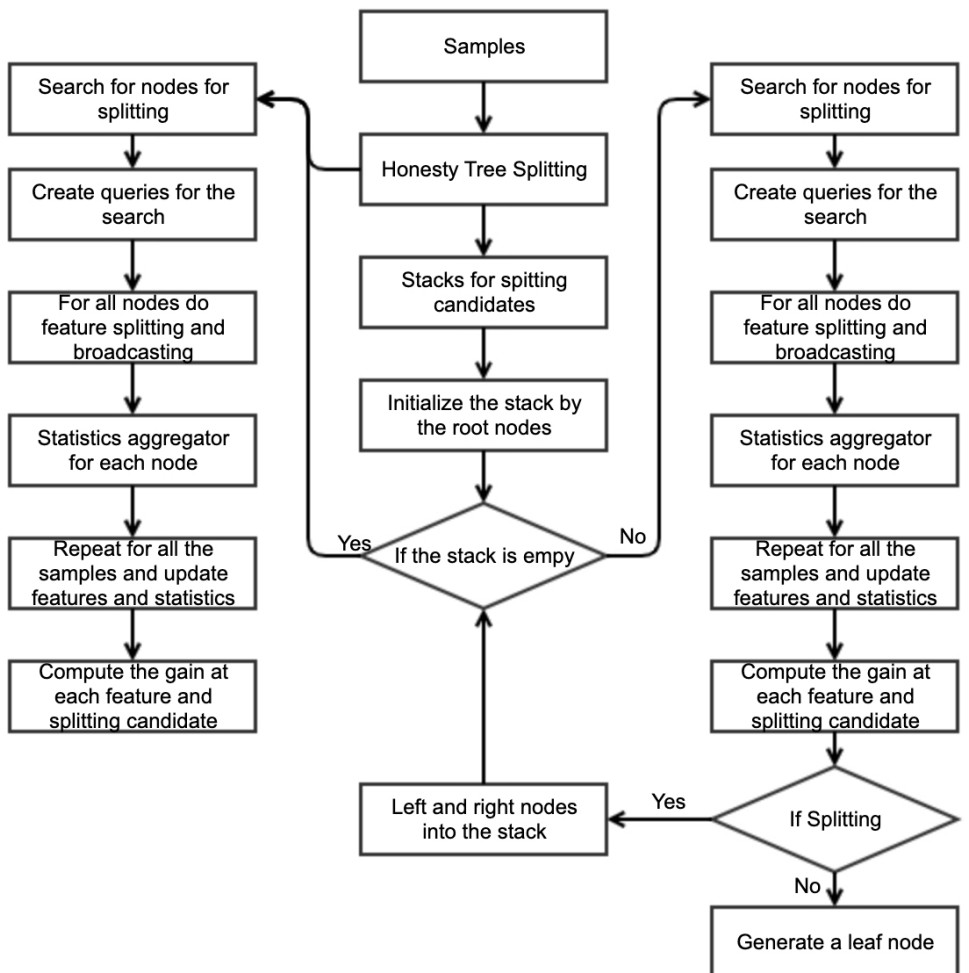

Figure 10: Diagram of GCF algorithm workflow

### 7.5    ONLINE DEPLOYMENT AND EXPERIMENTS

The deployment of models in the online marketplace is twofold as illustrated in Figure 11. On the one hand, the offline model that is trained on historical data provides HTE estimation that decision-makers need to customize incentives at the origin-destination-time level. On the other hand, the outcome in response to the customized incentives contributes to the data prepared for the subsequent model training. The following is an example of an online experiment with the GCF-driven policy.

We conduct online A/B testings with the model on 2 equal-size non-overlapping groups, i.e. the cohort. The evaluation metric of the online experiment is the finish order (FO) increment. The numerical results of GCF and CF are shown in Figure 12. GCF improves on CF by 15.1%, benefiting from the relaxation of the assumptions on the model specification. It shows that the more complicated or more general the treatment is, the more efficient the model is, which again shows the superiority of our novel combination of DRF and CF for continuous treatment.

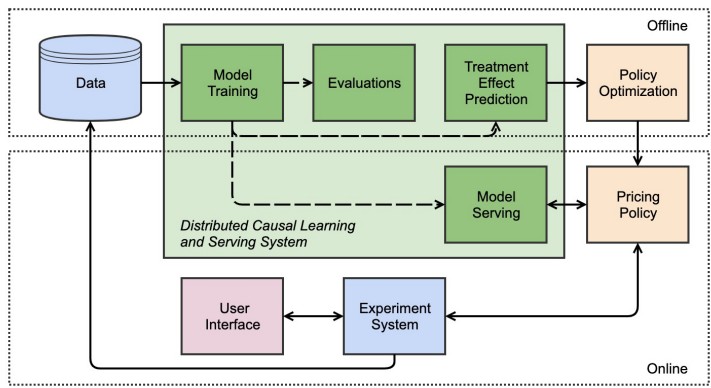

Figure 11: Online Experimental Design

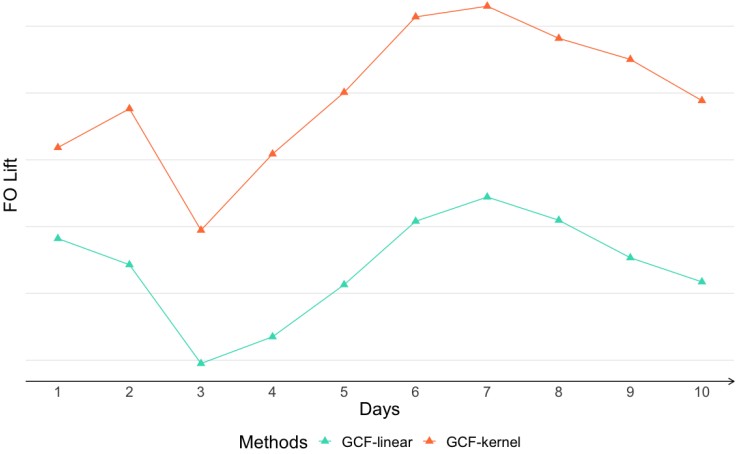

Figure 12: GCF vs CF on online experiment

