# OpenReview forum: "GCF: Generalized Causal Forest for Heterogeneous Treatment Effect Estimation Using Nonparametric Methods"
_ICLR.cc/2022/Conference — ICLR 2022 Submitted_

### Official Review · Reviewer_kG5h · 2021-10-29

**Correctness:** 3
**Technical Novelty And Significance:** 2
**Empirical Novelty And Significance:** 2
**Recommendation:** 5
**Confidence:** 3

**Main Review:**

I like the idea of generalizing the CF because it's impactful by allowing users to leverage any flexible modern machine learning tools. However, I think the paper is hardly understandable to most readers, and the theories are not deep enough.

Here are some details.
1. "However, these methods largely focus on binary ..."  may mislead, because this statement is only for a few recent estimators developed by Chernozhukov et al., Zhao et al., and Kunzel et al. For example, a BART estimator developed by Hill (2011) can deal with continuous treatments. I suggest conducting a more extensive literature review to provide more clear motivation.
2. What is the meaning of "local splitting" in the 3rd paragraph in Section 1? Please don't assume that the authors are familiar with the details of the causal forest algorithm.
3. This paper doesn't answer or motivate why we have to use the proposed GCF instead of the Kernel DML (Colangelo and Lee, 2020) or the doubly-robust estimator provided by Kennedy et al. (2017). It would be great if authors can provide more extensive literature review on the HTE estimators for the continuous treatment case.
4. In Section 2, what is the meaning of the "global" estimator?
5. I think the most important prerequisite to understanding the paper is the notion of the causal tree and causal forest, developed by Athey et al. Please add this in Section 3. Without it, it's indeed impossible to understand the paper. For example, what is the meaning of 'honesty' in Algorithm 1?
6. It's not clear what the contribution of the paper is. Even if I wrote the contribution as "replacing the partial linear model to the kernel DML," I only guessed so because it's not explicitly written.
7. I expected that the GCF would achieve the doubly robustness property, following the properties of the kernel DML. Please consider adding the formal convergence analysis exhibiting the doubly robustness property. Showing the empirical performance is not sufficient to promote the proposed estimator.
8. In the experiment, why don't you compare with the Kernel DML (Colangelo and Lee, 2020)?
9. There are typos -- "Table ??" on page 9, and "First, We" on page 2.

**Summary Of The Paper:**

This paper provides a generalized causal forest (CF) by replacing the partial linear model in the CF as a kernel DML estimator. Empirical evidences show that the proposed generalized CF (GCF) outperforms the existing estimators.

**Summary Of The Review:**

- This paper implicitly assumes that readers are familiar with the details of the causal forest algorithm. Without this prerequisite, it's impossible to understand the paper.

- This paper doesn't motivate to use the GCF over existing estimators (Kernel DML, or the one of Kennedy et al. (2017)).

- No theoretical analysis on the convergence or bias analysis for the GCF.

---

> ### Author Response · Authors · 2021-11-23
> **Response to Reviewer kG5h (Part 1)**
>
> > **Main Review:** I like the idea of generalizing the CF because it's impactful by allowing users to leverage any flexible modern machine learning tools. However, I think the paper is hardly understandable to most readers, and the theories are not deep enough.
> Here are some details.
> "However, these methods largely focus on binary ..." may mislead, because this statement is only for a few recent estimators developed by Chernozhukov et al., Zhao et al., and Kunzel et al. For example, a BART estimator developed by Hill (2011) can deal with continuous treatments. I suggest conducting a more extensive literature review to provide more clear motivation.
>
> Thanks so much for the suggestions! When we say 'these methods', BART is not included. But it's a great point for us to cover the relevant literature more inclusively. We have uploaded a new version of the paper with the rewritten literature review and BART is discussed. BART has the same problem with CF in that it requires a linear assumption.
>
> > What is the meaning of "local splitting" in the 3rd paragraph in Section 1? Please don't assume that the authors are familiar with the details of the causal forest algorithm.
>
> We apologize for jumping out of the terminology without a clear definition of it. The local splitting criterion is referred to as the criteria for growing a tree that partitions at each parent node to maximize the heterogeneity. Since the partition is at each parent node with respect to a subset of the feature space, it is named a local one.
>
> > This paper doesn't answer or motivate why we have to use the proposed GCF instead of the Kernel DML (Colangelo and Lee, 2020) or the doubly-robust estimator provided by Kennedy et al. (2017). It would be great if authors can provide more extensive literature review on the HTE estimators for the continuous treatment case.
>
> We would like to elaborate on the motivations and contributions of our paper. Firstly, compared to Kennedy, which derives the doubly robust estimator using the influence function, our doubly robust DRF estimator is derived from the score function by combining previous results from (Athey et al, 2019) and (Colangelo and Lee, 2020). Secondly, putting the kernel-based estimator into the node avoids the problem of the curse of dimensionality, which is a common concern for Kennedy's method.
> Not to be overshadowed, we are the first to implement the methods using parallel computing, which proves its fast running speed and high efficiency in the industry large-scale data. The comparison of running time on the real-world big dataset has been added to section 5.4 of the updated version.
>
> > In Section 2, what is the meaning of the "global" estimator?
>
> The global estimator is the one that computes the ATE by predicting with the regression model with all samples where all samples contribute equally. While a local one refers to the one given by the weighted average estimators. The estimator on each tree only takes into account neighborhood samples in the sense that they are on the same leaf node, which in turn leads to the different weights samples across the trees.
>
> > I think the most important prerequisite to understanding the paper is the notion of the causal tree and causal forest, developed by Athey et al. Please add this in Section 3. Without it, it's indeed impossible to understand the paper. For example, what is the meaning of 'honesty' in Algorithm 1?
>
> We have included the details on causal trees and the associated honesty principle in the related work of the paper. To have an overview of the honesty principle, when we grow a causal tree, we only utilize a subset of training samples. The remaining training samples only contribute to computing the ATE estimators on each leaf node of the tree at the prediction stage. There is no intersection between the two subsets. This principle is vital in two aspects. On the one hand, it deals with overfitting and thereby achieving bias reduction. On the other side, it leads to the convergence of the estimator.
>
> > It's not clear what the contribution of the paper is. Even if I wrote the contribution as "replacing the partial linear model to the kernel DML," I only guessed so because it's not explicitly written
>
> We would like to give more emphasis to the contributions of the paper and rewrite this part in Section 1. And your comment is precisely the highlight of this paper. We apply the more rubust kernel DML technique to the framework of causal forest as a solution to debias HTE estimation with non-parametric modeling when given data with more complexity and to better handle high-dimensional features. The implementation of the proposed methods in parallel computing is also crucial for the application in large-scale data, which is common in the field of e-commerce, online marketplace.

---

> ### Author Response · Authors · 2021-11-23
> **Response to Reviewer kG5h (Part 2)**
>
> > I expected that the GCF would achieve the doubly robustness property, following the properties of the kernel DML. Please consider adding the formal convergence analysis exhibiting the doubly robustness property. Showing the empirical performance is not sufficient to promote the proposed estimator.
>
> The double robustness of the method shall be guaranteed by the asymptotic convergence of CF and the double robustness of the kernel-based DML estimator. We have included a proof sketch in Section 4 by showing that the assumptions are indeed valid and we appreciate your reading that part.
>
> > In the experiment, why don't you compare with the Kernel DML (Colangelo and Lee, 2020)?
>
> The code implementation we used here of Kennedy's methods is adapted from the author's original code(https://github.com/ehkennedy/npcausal/blob/master/R/ctseff.R).
> In this implementation, we borrow the model from the SuperLearner package as the base learner to construct Kennedy's doubly robust estimator, which already includes a cross-validation mechanism in the code. Therefore,  implementing Kennedy's method is already a good representative for kernel DML in the sense that they should have similar numerical performance.
>
> > There are typos -- "Table ??" on page 9, and "First, We" on page 2.
>
> We have fixed the typos in the new version.
>
> > **Summary Of The Review:** This paper implicitly assumes that readers are familiar with the details of the causal forest algorithm. Without this prerequisite, it's impossible to understand the paper.
> This paper doesn't motivate to use the GCF over existing estimators (Kernel DML, or the one of Kennedy et al. (2017)).
> No theoretical analysis on the convergence or bias analysis for the GCF.
>
> Dear Reviewer, thanks so much for your time and effort into this paper! These comments mean a lot to us and help us to perceive the contents of the paper in many ways. In the updated version of the paper, we have included more details about causal forest in the second paragraph of related work, such as the tree splitting and the honesty principle. Meanwhile, we rewrite the contributions in the introduction to have a better presentation and concentrated points of the paper. In summary, our generalization is important since it avoids the curse of dimensionality suffered by (Kennedy et al. (2017), kernel DML) by adopting a forest framework and gets rid of the limitations of linear or parametric assumptions in Athey et.al (2019). And we provide a proof sketch by checking the validity of assumptions of CF under the assumptions of kernel DML estimators and a lipschitz condition. In the practical sense, we implement the algorithm on Spark to enable distributed and parallel computing which reach an approximately 50% reduction in running time.  We would appreicate your revisiting the responses and the updated paper.

---

> > ### Comment · Reviewer_kG5h · 2021-11-25
> > **Response**
> >
> > ## Other note
> > After re-reading the paper, I found that this paper's result is novel in that it provides an HTE (Heterogeneous treatment effect) estimator where the treatments are continuous. To my understanding, [Colangelo and Lee, 2020] only cover the ATE, not the HTE. So, I think this paper can be differentiated.
> >
> > For more note, a list of recent HTE estimators that leverages modern ML models is {T, S, X}-learner [Kunzel et al., 2019], R-learner [Nie and Wager, 2020], and DR-learner [Kennedy, 2020], to cite a few. These are either only considering the binary treatment or non-doubly robust. I think the authors' proposed method is novel in that it offers a doubly robust HTE estimator that can cover continuous treatment.
> >
> > ## On some responses
> > > The double robustness of the method shall be guaranteed by the asymptotic convergence of CF and the double robustness of the kernel-based DML estimator. We have included a proof sketch in Section 4 by showing that the assumptions are indeed valid and we appreciate your reading that part.
> >
> > Given that the paper provides a theoretically novel estimator, I encourage to unfold Section 4.4 and claim a formal convergence result as a Theorem. For example, see [Thm.1 of Kunzel et al., 2019] or [Thm. 2 of Kennedy, 2020].
> >
> > Overall, I think the result is significant in theoretically and empirically. However, the paper can be improved by
> > * adding more preliminaries on the causal forest algorithm,
> > * providing formal statistical convergence rate analysis, and
> > * providing extensive literature review on HTE estimators & continuous treatment cases.
> >
> > P.S., In Section 4.4, I think unbiased -> consistency. Please check.

---

> > > ### Author Response · Authors · 2021-11-27
> > > **Response**
> > >
> > > > **Other note**
> > > After re-reading the paper, I found that this paper's result is novel in that it provides an HTE (Heterogeneous treatment effect) estimator where the treatments are continuous. To my understanding, [Colangelo and Lee, 2020] only cover the ATE, not the HTE. So, I think this paper can be differentiated.
> > >
> > > >For more note, a list of recent HTE estimators that leverages modern ML models is {T, S, X}-learner [Kunzel et al., 2019], R-learner [Nie and Wager, 2020], and DR-learner [Kennedy, 2020], to cite a few. These are either only considering the binary treatment or non-doubly robust. I think the authors' proposed method is novel in that it offers a doubly robust HTE estimator that can cover continuous treatment.
> > >
> > > Dear reviewer, thanks so much for your patience and time! We very much appreciate your instructions and suggestions for this paper.
> > >
> > > The essence of this paper is solving **continuous treatment effect** using **machine learning models**. In fact, in the early version of this paper, we emphasize more on the ML models by comparing our methods to the current popular ML models you just mentioned. This narrative put these ML models in an unfair position as our simulated treatment effects are all continuous. This consideration drives us to re-write the literature review to emphasize more on the **continuous treatment effect** which is the version you're reviewing now.
> > >
> > > Nevertheless, we agree with your comments that discussions on these models should be part of the literature review. We will add them to our future version.

---

> > > > ### Author Response · Authors · 2021-11-27
> > > > **Response (cont.)**
> > > >
> > > > > **On some responses**
> > > > The double robustness of the method shall be guaranteed by the asymptotic convergence of CF and the double robustness of the kernel-based DML estimator. We have included a proof sketch in Section 4 by showing that the assumptions are indeed valid and we appreciate your reading that part.
> > > >
> > > > > Given that the paper provides a theoretically novel estimator, I encourage to unfold Section 4.4 and claim a formal convergence result as a Theorem. For example, see [Thm.1 of Kunzel et al., 2019] or [Thm. 2 of Kennedy, 2020].
> > > >
> > > > Thanks for providing the great reference! We will work on formalizing the statement by summarizing a theorem with proof in Section 4.4.
> > > >
> > > > > Overall, I think the result is significant in theoretically and empirically. However, the paper can be improved by
> > > > adding more preliminaries on the causal forest algorithm,
> > > > providing formal statistical convergence rate analysis, and
> > > > providing extensive literature review on HTE estimators & continuous treatment cases.
> > > >
> > > > Dear reviewer, your professional suggestions and the acknowledgment of the paper are of great importance to our research. We would like to show our great gratitude and revise the paper by following the instructions.
> > > >
> > > > * Add details on Causal Forest such as honesty principle, construction of score function in Section 3.
> > > > * Substitute the statements in Section 4.4 with a theorem and provide complete convergence analysis under explicit assumptions.
> > > > * Include more literature related to HTE estimation with continuous treatments in addition to the existing ML models.
> > > >
> > > > > P.S., In Section 4.4, I think unbiased -> consistency. Please check.
> > > >
> > > > We apologize for the abuse use of the word 'unbiased' instead of the precise one as consistency or asymptotically unbiased while characterizing the convergence behavior. We will fix it in the later version.

---

### Official Review · Reviewer_X6oe · 2021-11-03

**Correctness:** 1
**Technical Novelty And Significance:** 1
**Empirical Novelty And Significance:** Not applicable
**Recommendation:** 1
**Confidence:** 4

**Main Review:**

Weaknesses:
- The paper does not contain new methodology; it is purely a combination of existing techniques (generalized random forest and doubly robust estimator).
- The paper does not provide a theoretical analysis of the proposed method.
- The authors oversell their contribution: Generalizing the Generalized Random Forest (Athey et al. 2019) from a partially lienar model to a full nonparametric one is not very creative.
- It is unclear to me how the estimator $\hat{\mu}(t, X_i)$ is constructed. It is the weighted average of something (?), where the weights w_i(x) are presumably the frequency with which the ith training example falls into the the same leaf as x (see Athey et al. 2019).
- The simulation study is not reproducible because the authors do not include values for sample size $n$ and dimension $p$.
- The simulation study and real data analysis are very small scale.
- Unsupported claims: In the introduction the authors claim that their method allowed for faster hyper-parameter tuning than competing methods. However, they do not provide any evidence to back-up this claim.

Additional comments:
- Poor notation: For example, on page 3 $\Omega$ is defined as a single vector, on page 4 the authors draw samples from $\Omega$, in Algorithm 1 on page 5, $\Omega$ denotes the dat set.
- The exposition is unorganized and contains many abbreviations which are hard to keep track of.
- The language is unnecessarily pompous and riddled with grammatical mistakes to an extend that makes it difficult to follow the authors ideas.

**Summary Of The Paper:**

The authors extend the generalized random forest by Athey et al. (2019) from partially linear models to nonparametric ones and combine it with a doubly robust estimation step. The paper contains a small scale simulation study and a small real data analysis.

**Summary Of The Review:**

The paper does not contain new methodology or theoretical result. The simulation study and real data analysis are weak.

---

> ### Author Response · Authors · 2021-11-23
> **Reply to Reviewer X6oe (Part 2)**
>
> > Additional comments:
> Poor notation:
> For example, on page 3 Ω is defined as a single vector, on page 4 the authors draw samples from Ω, in Algorithm 1 on page 5, Ω denotes the dat set.
> The exposition is unorganized and contains many abbreviations which are hard to keep track of.
> The language is unnecessarily pompous and riddled with grammatical mistakes to an extend that makes it difficult to follow the authors ideas.
>
> >**Summary Of The Review:** The paper does not contain new methodology or theoretical result. The simulation study and real data analysis are weak.
>
> In this paper, we propose a new algorithm by combining kernel DML estimators with Causal Forest that can get rid of the limitations of both counterparts. We can model the non-linear or non-parametric relationship between responses and treatments by non-parametric splitting criterion and as well as avoiding the curse of dimensionality by means of causal forest compared to the non-parametric estimators.  We also include a demonstration on the convergence property of this new estimator by revisiting the conditions and theorems in Athey et.al (2019) and kernel DML.  In the meantime, we implement the proposed algorithm with Spark and achieve 50% reduction in running time. We show the empirical validity and efficiency of GCF on both synthetic datasets that perfectly show the complex non-linear relationships between the response and treatments and large-scale real-world datasets with size more than10 million. Moreover, we would like to refer the reviewer for results on an online A/B testing by deploying GCF and compteing methods which again show the efficiency of this new method.

---

> ### Author Response · Authors · 2021-11-23
> **Reply to Reviewer X6oe (Part 1)**
>
> > **Summary Of The Paper:** The authors extend the generalized random forest by Athey et al. (2019) from partially linear models to nonparametric ones and combine it with a doubly robust estimation step. The paper contains a small scale simulation study and a small real data analysis.
>
> Thanks for your time and the comments, dear Reviewer. Please find the responses to your detailed comments below.
>
> > **Weaknesses:**
> The paper does not contain new methodology; it is purely a combination of existing techniques (generalized random forest and doubly robust estimator).
> The paper does not provide a theoretical analysis of the proposed method.
> The authors oversell their contribution: Generalizing the Generalized Random Forest (Athey et al. 2019) from a partially lienar model to a full nonparametric one is not very creative.
>
> The generalization of Causal Forest is two-phase. One side is that our proposed method can fit into the non-parametric modeling by adapt DRF to the splitting criterion. A linear one can only utilize the slope by regressiing Y over T, which may neither capture the local behavior of treatments such as concave functions, nor utilize various distance metrics. On the other side, the more rubust DML estimator is embeded into the splitting criterion and thereby dealing with curse of dimensionality and reducing bias through weighting average and bagging. And our Spark implementation of our method guarantees the effectiveness of the idea of combining these two is guaranteed and thereby leading to more practical use.
>
> > It is unclear to me how the estimator $\mu(t,X_i)$ is constructed. It is the weighted average of something (?), where the weights $w_i(x)$ are presumably the frequency with which the ith training example falls into the the same leaf as x (see Athey et al. 2019).
>
> The construction of the causal estimand is two-fold. First, the in-node estimator is derived from the kernel-based DML estimator as described in sec 4.2. Then we average all estimands across nodes. The weight is determined by the sample in each node. We have added detailed descriptions in the algorithm pseudocode in our newly updated paper.
>
> > The simulation study is not reproducible because the authors do not include values for sample size n and dimension p.
> The simulation study and real data analysis are very small scale.
>
> For the simulation study, the sample size is 1000 and dimension p is 50 and we have put the setup in the experimental section. This toy example coincides with the statement that our algorithm is applicable to nonlinear model inference, since real-world datasets may not perfectly meet the condition of linearity. We further show the validity of this algorithm on real-world datasets with more than 10 million data entries. The scope of this numerical experiment is large and our algorithms performs the best compared to competing methods. We add these necessary details to the experimental section (Sec 5.3.2) and would appreciate your kindly reading that part.
>
> > Unsupported claims: In the introduction the authors claim that their method allowed for faster hyper-parameter tuning than competing methods. However, they do not provide any evidence to back-up this claim.
>
> Our algorithm is implemented on Spark that allows for distributed computing. Our implementation significantly reduces the running time by more than 50% (~1hr -> ~30min) compared to the benchmark (Causal Forest, implemented in C++) in the real-world dataset of more than 10M entries. This is what supports faster hyper-parameter tuning. We apologize for not including the running time for each method.

---

> ### Author Response · Authors · 2021-11-29
> **We would like to hear from you**
>
> Dear Reviewer X6oe,
>
> Thanks again for your comments and suggestions for our paper. We are more than happy to answer any further concerns you may have. Please do not hesitate to comment and let us know your feedback.
>
> Thank you for your time!
>
> Paper author

---

### Official Review · Reviewer_akBL · 2021-11-06

**Correctness:** 3
**Technical Novelty And Significance:** 2
**Empirical Novelty And Significance:** 3
**Recommendation:** 5
**Confidence:** 2

**Main Review:**

Strength:

1. The paper is concerned with an interesting and important problem in causal inference, which is heterogeneous treatment effect estimation with continuous treatment. Modeling the nonlinearity of the dose-response function is also an important and challenging aspect of this problem.

2. The paper presents solid empirical results on real-world data. The paper also presents a real-world a/b test to demonstrate the effectiveness of the proposed method in practice.

3. The paper also presents a scalable implementation of the proposed method, which can be used to handle large-scale (production-scale) data.

Weakness:
1. Presentation is not clear. Important concepts such as score function, splitting criteria, etc are not specified. The presentation of the proposed method is also difficult to follow. For example, it is not clear to me what are the components of CF, and how GCF is derived and differs from CF in section 4.

2. The key component of the proposed method is to model the DRF in a non-parametric, doubly robust fashion. It is not clear to me whether such a contribution is significant enough to motivate acceptance. This is also because the proposed method performs comparably with Kenndy on synthetic data.  So, it is not clear what might be the additional value brought compared to Kenndy. The author may consider providing further explanation and justification on the complexity and technical difficulty of replacing a linear DRF with the proposed one.

3. There is a lack of theoretical justification of the proposed method. However, the empirical results on synthetic data and real-world experiments help to assuage this concern.

**Summary Of The Paper:**

The paper proposes a generalized version of the causal forest for heterogeneous treatment effect estimation for continuous treatment by non-parametric modeling of the dose-response function. To provide non-parametric modeling, the proposed method makes use of kernel-based double/debiased estimators. Experiments of the proposed methods are run on assorted synthetic datasets as well as real-world datasets to demonstrate the effectiveness of the proposed method compared to competing methods.

**Summary Of The Review:**

Correctness: I did not identify major issues in terms of correctness.
Novelty and significance: I have concerns about the significance of the proposed method in terms of modeling the RDF through non-parametric approaches. I appreciate the authors running real-world experiments with a/b test to demonstrate the performance of the proposed method.

---

> ### Author Response · Authors · 2021-11-23
> **Reply to Reviewer akBL**
>
> >**Weakness:** Presentation is not clear. Important concepts such as score function, splitting criteria, etc are not specified. The presentation of the proposed method is also difficult to follow. For example, it is not clear to me what are the components of CF, and how GCF is derived and differs from CF in section 4.
>
> The basic component of causal forest is random forest with treatment effect estimators.
>
> Its building block relies on a splitting criterion that partitions the feature space of parent nodes into those of child nodes for dimensionality reduction.
>
> In the context of treatment effect estimations, the splitting criterion is the difference of a metric between child nodes that measures to what degree the response varies with treatments, which can be estimated via score functions. The partition aims at maximizing the corresponding splitting criterion. Thus far, we could imagine that the splitting criterion is at the heart of causal forest, which is the departure of our GCF from CF.
>
> We generalize the partially linear model of CF which only allows for linear-type relationship between the response and treatments to a non-parametric one by borrowing the concept of dose-response functions. Based on that, we construct a new distance-based splitting criterion with partial DRF plugged in where DRF is obtained from kernel-based DML estimators.
>
> This new splitting criterion can better capture the non-linear or even non-parametric relationship between the response and treatments. We hope that the clarifications would be of any help for conveying the key idea of the paper.
>
> > The key component of the proposed method is to model the DRF in a non-parametric, doubly robust fashion. It is not clear to me whether such a contribution is significant enough to motivate acceptance. This is also because the proposed method performs comparably with Kenndy on synthetic data. So, it is not clear what might be the additional value brought compared to Kenndy. The author may consider providing further explanation and justification on the complexity and technical difficulty of replacing a linear DRF with the proposed one.
>
> We would like to elaborate on the motivations and contributions of our paper. Firstly, compared to Kennedy, which derives the doubly robust estimator using the influence function, our doubly robust DRF estimator is derived from the score function by combining previous results from (Athey et al, 2019) and (Colangelo and Lee, 2020). Secondly, putting the kernel-based estimator into the node avoids the problem of the curse of dimensionality, which is a common concern for Kennedy's method.
> Not to be overshadowed, we are the first to implement the methods using parallel computing, which proves its fast running speed and high efficiency in the industry large-scale data. The comparison of running time on the real-world big dataset has been added to section 5.4 of the updated version.
>
> > There is a lack of theoretical justification of the proposed method. However, the empirical results on synthetic data and real-world experiments help to assuage this concern.
>
> We haven't yet fully analyzed the theoretical proof of the method, but instead provided a proof sketch of the asymptotic property of our estimators under slightly stronger assumptions in Section 4.4. We would like to refer the reviewer there for a demonstration of the asymptotic property and will give efforts to provide the full proof at the next stage.
>
> > **Summary Of The Review:**
> Correctness: I did not identify major issues in terms of correctness.
> Novelty and significance: I have concerns about the significance of the proposed method in terms of modeling the RDF through non-parametric approaches. I appreciate the authors running real-world experiments with a/b test to demonstrate the performance of the proposed method.
>
> Great point and thanks a lot for the reminder! We present the results of our proposed GCF on an online A/B experiment in the Appendix and more details should be found there out of page limit. Overall, our algorithm achieves the best Qini Score and the highest KPI compared to several widely-used baselines. We would appreciate it if the reviewer could take any time to review Section 7.5 which is in Appendix.

---

> ### Author Response · Authors · 2021-11-29
> **We would like to hear from you**
>
> Dear Reviewer akBL,
>
> Thanks again for your comments and suggestions for our paper.
> We are more than happy to answer any further concerns you may have. Please do not hesitate to comment and let us know your feedback.
>
> Thank you for your time!
>
> Paper author

---

### Decision · Program_Chairs · 2022-01-20

**Decision:**

Reject

**Comment:**

The paper introduces some interesting ideas on how use causal random forests for conditional average treatment effects (CATE), with respect to some baseline treatment level ("0"), when the treatment variable is continuous. Figure 1 summarises the scope of the paper neatly. Scalability issues are also considered.

I think this *is* a paper "nearly there" in terms of a impactful contribution. The main issues are some presentation kinks and extra steps in the theory. I think the very low scores from the reviewers are not quite representative of the overall quality (I would be more generous). However, I'm afraid I'm also inclined towards a reject. The paper neglects some other developments on ML for CATE with continuous treatment e.g. Bica et al.'s "Estimating the Effects of Continuous-valued Interventions using Generative Adversarial Networks" (NeurIPS 2020) and the references within. A focus on the theory would help to differentiate it, but I'm not that confident that the results are currently mature enough to claim them.

Although I suggest a rejection, let me make clear I strongly encourage the authors to further pursue their ideas. You are doing good work, and the next iteration might nail it. As you found out in the discussion, emphasise the continuous aspect of it. I'd also emphasise the fact that you have a clear setup of the problem in terms of the contrast wrt to a baseline treatment effect instead of some generic contrast function. People in ML tend to be oblivious to such a setup, but I'm not convinced you are properly exploiting it.